# Ammonification by kelp associated microbes increases ammonium availability

**Alex Hochroth** [1] *, **Catherine A. Pfister** [2,3]

1 The College, The University of Chicago, Chicago, IL, United States of America, 2 Committee on Evolutionary Biology, The University of Chicago, Chicago, IL, United States of America, 3 Department of Ecology & Evolution, The University of Chicago, Chicago, IL, United States of America

* alexhochroth@gmail.com

**Data Availability Statement:** The metagenome data are published are available at NCBI's Sequence Read Archive, accession no. PRJNA783443. The minimal dataset underlying the results can be found at Figshare: https://figshare.

## Abstract

Microbes contribute biologically available nitrogen to the ocean by fixing nitrogen gas from the atmosphere and by mineralizing organic nitrogen into bioavailable dissolved inorganic nitrogen (DIN). Although the large concentration of plants and algae in marine coastal environments provides ample habitat and reliable resources for microbial communities, the role of the microbiome in host-microbe nitrogen cycling remains poorly understood. We tested whether ammonification by epiphytic microbes increased water column ammonium and improved host access to nitrogen resources by converting organic nitrogen into inorganic nitrogen that is available for assimilation by hosts. When bull kelp (*Nereocystis luetkeana*) in the northeast Pacific was incubated with $^{15}N$ labelled amino acid tracers, there was accumulation of $^{15}N$ in kelp tissue, as well as accumulation of $^{15}NH_4$ in seawater, all consistent with the conversion of dissolved organic nitrogen to ammonium. Metagenomic analysis of surface microbes from two populations of *Nereocystis* indicated relative similarity in the percentage of genes related to ammonification between the two locations, though the stressed kelp population that had lower tissue nitrogen and a sparser microbiome had greater ammonification rates. Microbial communities on coastal macrophytes may contribute to the nitrogen requirements of their hosts through metabolisms that make ammonium available.

## Introduction

Nutrients, including nitrogen, can limit plant productivity across diverse ecosystems [1], including the coastal ocean [2]. Where nitrogen is limiting in the coastal ocean, there may be selection for macrophytes to have microbial associations that assist in acquiring nitrogen, such as the association of nitrogen-fixing bacteria [3–6]. In addition, microbial associations may also increase access to nitrogen through increased microbial transformation of organic nitrogen. Ammonification, the metabolism of amino acids into ammonium, is a ubiquitous microbial metabolism [7]. Enzymes such as hydrolases and lyases cleave the amide group of some amino acids to release dissolved inorganic nitrogen (DIN) in the form of ammonium [8]. Amino acids and other forms of dissolved organic nitrogen (DON), including urea, nucleic acids, proteins, and peptides, can be important sources of organic nitrogen. These DON

com/projects/Plos_One_Nereocystis_
Ammonification/180562. All other relevant data are
within the manuscript and its Supporting
Information files.

**Funding:** Funding was provided by an Ecology and
Evolution Undergraduate Research Fellowship
from UChicago (to AH) and Washington
Department of Natural Resources #93100399 (to
CAP). University of Chicago Ecology and Evolution:
https://ecologyandevolution.uchicago.edu/
Washington Department of Natural Resources:
https://www.dnr.wa.gov/ The funders had no role
in study design, data collection and analysis,
decision to publish, or preparation of the
manuscript.

**Competing interests:** The authors have declared
that no competing interests exist.

components can have rapid turnover [9,10], and their concentrations are greatly affected by the metabolic activities of animals and macrophytes [11,12]. Nitrogen in the form of DON may be largely inaccessible to macrophytes until it is converted to DIN by microbial metabolisms. While phytoplankton have been shown to have extracellular enzymes that cleave amino acids [11,13], there is not yet a corresponding demonstration in algae or seagrass. Though previous studies suggest that macrophytes may be able to directly take up urea as a source of organic nitrogen [14,15], most of these studies do not control for or investigate the presence of host-associated bacteria that may facilitate that uptake. In fact, it was demonstrated that microbial amino acid ammonification provisioned a seagrass species with released ammonium when the bacterial community was investigated through imaging and stable isotope enrichments [16] and through in situ isotope incubations [17]. This suggests that epiphytic microbial communities may play a vital role in making DON available to the host.

Sources of DON can be abundant and reach concentrations that rival the total amount of DIN [18]. Though amino acid concentrations in the coastal oceans are not well studied, some estimates put them as high as 2 μmol/L in coastal areas, much greater than the 0.2 μmol/L concentrations estimated to exist in the open ocean [19]. This indicates that DON could represent a significant potential source of nitrogen for microbes in coastal environments and could select for specific host-microbe interactions. Macrophytes host diverse microbial taxa that can use dissolved organic matter [6,20,21], though little is known about any benefits those microbes provide. While amplicon-based and metagenomic studies increasingly capture the microbial diversity of marine macrophytes [22–24], we know comparatively little about the functions of microbial diversity and how they relate to the condition of the host.

We quantified ammonification by microbes on host seaweeds and tested whether this ammonification increased nitrogen availability to their hosts. *Nereocystis luetkeana* (henceforth *Nereocystis)* blades have been shown to host microbes on their surfaces with enzymes for ammonification [6]. We quantified microbial transformation and host nitrogen uptake using stable isotope enrichments in the bull kelp *Nereocystis* across three locales in the northeast Pacific.

We quantified ammonification in *Nereocystis* samples collected from two disparate geographic areas that differ in microbial diversity, microbial abundance, and kelp bed growth and density. Microbial communities on *Nereocystis* exhibit strong geographic differences that correlate with the health of the host kelp population. CLASI-FISH imaging [25], and 16S gene sequencing [20] were used to quantify microbial diversity and abundance on *Nereocystis* in South Puget Sound and the outer coast of Washington State. Kelp-associated microbes in South Puget Sound, where kelp populations have been in decline [26], show decreased diversity and abundance compared to communities on the outer coast of Washington, where kelp flourish [27]. Further analysis of bacterial metagenome-assembled genomes (MAGS) from the persistent versus declining population of *Nereocystis* [21] allowed us to determine if ammonification genes were present in the microbiome of these hosts and whether the incidence and prevalence of these genes differed among the sites. Our results provide evidence that ammonifying microbes are present in association with hosts and in surrounding seawater; their activities may contribute to the nitrogen requirements for *Nereocystis* across several different geographic locations, broadening our understanding of the sources of nitrogen available to foundational coastal organisms.

## Methods

### Study sites

We studied *Nereocystis* across a large geographic gradient that encompassed a healthy and persistent population at Tatoosh Island (48.393689, -124.733820) [27] on the outer coast of

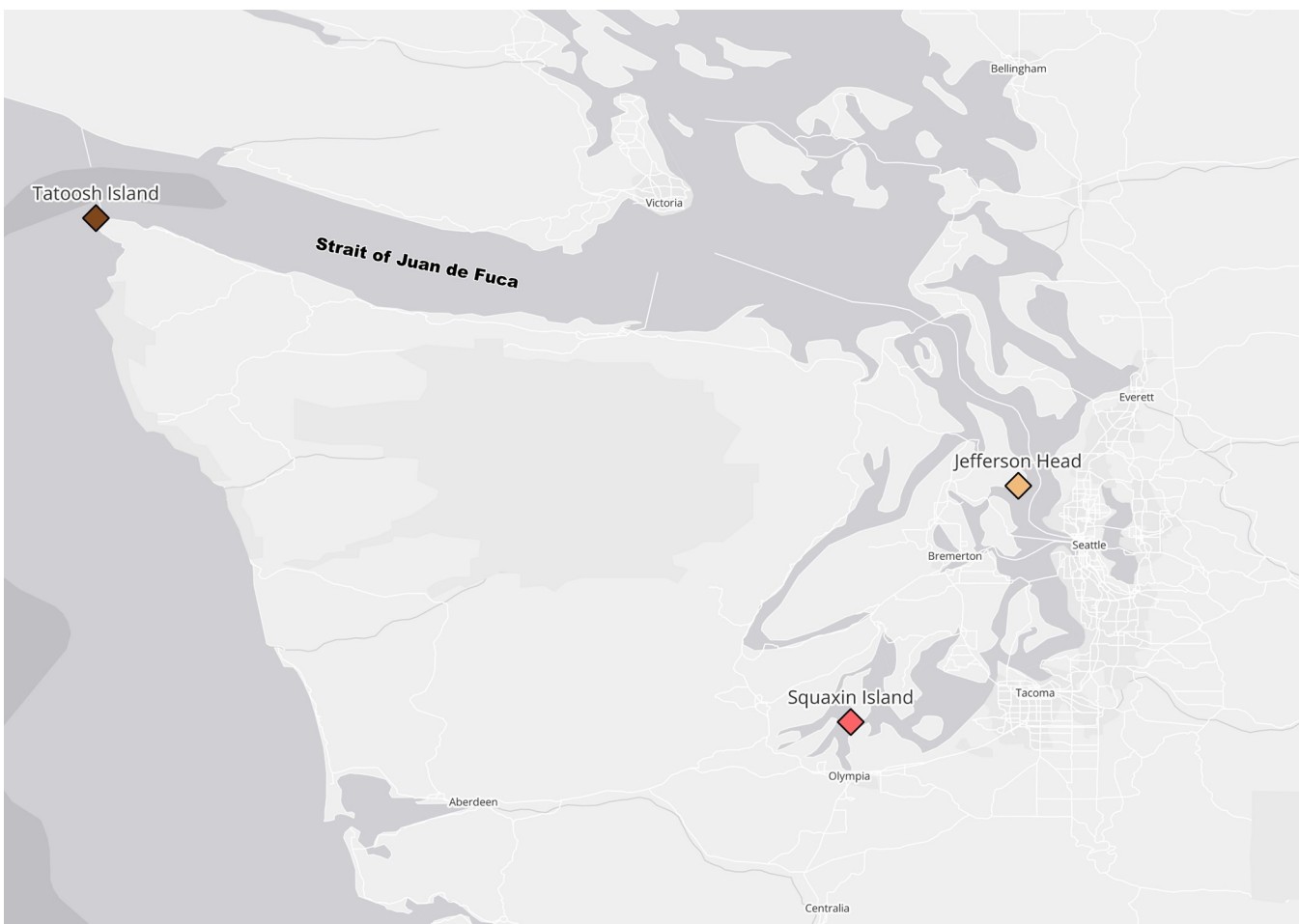

**Fig 1. Sampling locations.** Locations of bull kelp incubations in Washington State. Tatoosh Island is in the western Strait of Juan de Fuca, while Jefferson Head and Squaxin Island are in Puget Sound. Each set of incubations included n = 2 seawater-only chambers as a control, and n = 4 chambers containing kelp and seawater. Map made using data from Natural Earth.

Washington State, a declining population at Squaxin Island in Puget Sound (47.167282, -122.895984) [26], and a newly restored community in Puget Sound at Jefferson Head (47.742128, -122.488112) which was historically a kelp bed; now, *Nereocystis* are outplanted on line substrates. All sampling locations are shown in Fig 1.

## Chambers and incubation

We quantified ammonification and ammonium uptake using short term (three hour) chamber incubations with $^{15}N$-enriched amino acids. Experimental incubation time was informed by Weigel and Pfister [28], such that the $^{15}N$ signal had enough time to move through the system, but not long enough for nutrient depletion to occur. In total, there were three sets of incubations from July 2021 to August 2021. Kelp blade samples were placed in custom-made chambers as described in Weigel & Pfister [28]. Kelp chambers were 2.6L polycarbonate tubes capped at each end with a wing-nut expansion plug and a rubber seal, which ensured the tubes were watertight and easy to open for sampling.

Entire non-reproductive bull kelp blades were collected by boat or from the surface at low tide. Wet mass of macrophytes was measured with a Pesola™ (Forestry-Suppliers Inc.,

Jackson Mississippi, USA) spring scale; dry mass was measured after samples were dried for 48h at 50°C. Bull kelp blade dry mass averaged 3.12 ± 0.4g.

Kelp tissue was placed individually in each of 4 chambers and filled entirely with fresh seawater collected from the same location and depth as the samples; bottles or chambers filled with only seawater ($n = 2$) for each experiment served as a control for microbial activity in the water column. At time $T_0$, we added 200μL of 0.025M $^{15}$N-labeled amino acid solution (Cambridge Isotope Labs, product# NLM-2161, Lot# PR-24163) to the kelp chambers. The enriched amino acid mix was composed of 16 amino acids, with leucine, glutamic acid, and alanine making up 37% of the total. The amino acids valine, phenylalanine, isoleucine, aspartic acid, glycine, threonine, proline, tyrosine, arginine, lysine, serine, methionine, and histadine all composed between 2% and 9% of the mix. Amino acid concentrations are rarely estimated in coastal areas, and we do not have measurements in the locales where our experiments were performed. Instead, we assumed a concentration of 1 μM amino acids in seawater, based on previous ocean studies [29–31]. Our enrichment achieved 525,430‰ $^{15}$N, compared to the natural amount of $^{15}$N that we are assuming is present in seawater with a concentration of 1 μM amino acids. This high enrichment ensured the concentration of $^{15}$N that was added to the seawater served as an effective tracer, while only elevating amino acid concentrations to approximately 1.92 μM. The chambers were sealed and agitated by hand to mix the tracer evenly with the seawater. We suspended kelp chambers horizontally in a recreational float that kept them immersed in the water. This allowed temperatures in the chambers and bottles to remain close to ambient seawater temperature and permitted gentle water movement. Sampling occurred only at the beginning and end of the experiment to avoid contamination among chambers. The theorized movement of enriched $^{15}$N through the closed system is illustrated in S1 Fig.

## Measuring nutrients and the fate of tracer $^{15}$N

We measured seawater nutrient concentrations and $\delta^{15}NH_4$ at the beginning and end of the experiment by collecting two water samples from the source water we used before the addition of the tracer ($T_0$), and then again from each individual container ($T_f$). We measured concentrations of ammonia, nitrate, nitrite, phosphorous, silica, and $\delta^{15}NH_4$. For both samples we filtered 25mL and 50mL of seawater, respectively, through a syringe filter (Whatman GF/F, 0.7 μm) into acid-washed HDPE (high density polyethylene) bottles. All seawater samples were frozen for later analysis. All nutrient concentrations were analyzed at the University of Washington Marine Chemistry Laboratory (methods from UNESCO [32]), while seawater isotope determinations were done at the University of California, Davis. To quantify the isotopes of nitrogen in ammonium, we needed higher concentrations of $^{15}NH_4$ in seawater than were typical at the sites. We added 77.6μL of 0.05 M $NH_4Cl$, effectively adding 1.4 mg/L of $NH_4$ to all samples prior to analysis. The $\delta^{15}NH_4$ value of the $NH_4Cl$ was known (-2.11‰) and was a consistent addition to all samples.

Tissue was sampled from paired bull kelp blades to quantify percent carbon and nitrogen analysis, as well as the $\delta^{15}$N of incubated tissues at $T_0$ and at $T_f$. We collected the rapidly growing tissue at the basal meristem. Samples were weighed and analyzed on an elemental-analyzer–isotope-ratio mass spectrometer at Northwestern University Stable Isotope Biogeochemistry Laboratory (NUSIBL).

## Quantifying microbial transformations and macrophyte uptake

We used stable isotope tracers to quantify the mineralization of amino acids into $^{15}NH_4$ by following the transfer of the tracer from its source (the labeled amino acids) to its sink (the

seawater), modeling the process according to Lipschultz [33]. This model estimates a single rate parameter from $T_0$ to $T_f$ using the following difference equation:

$$Ammonification\ Rate = \frac{R(t)_{sink} - R(0)_{sink}}{(R_{source} - R(0)_{sink}) * \Delta t} * \overline{[NH_4]} \tag{1}$$

where $R$ denotes the isotopic ratio of the sink or source in atom%, $\Delta t$ represents the length of the incubation, and $\overline{[NH_4]}$ represents the average concentration of ammonium over the course of the experiment. This equation quantifies the transfer of $^{15}N$ from amino acids to ammonium in the water column. This version of the model approximates the rate of $^{15}N$ transfer as the difference between the isotopic ratio at the beginning of the incubation and the isotopic ratio at the end of the incubation, when our two samples were collected. Multiplying by the average ammonium concentration yields the uptake rate of ammonium by seawater and accounts for the rapid flux of inorganic nitrogen [34]. We assumed that sources of $^{15}N$ in the chambers were negligible compared to the tracer ($10^{-6}$M vs. 0.025M) and anticipated that isotope dilution due to mineralization of unlabeled DON would not be an important factor with an incubation of this duration.

We estimated ammonium uptake rates in kelp from the $^{15}N$ concentrations within the tissue, using the method for estimating nitrogen uptake from isotope enrichment described in Pather et al [35]. This method uses the following equation:

$$NH_4^+\ uptake\ rate = \frac{tissue\ 15N\ atom\%\ excess}{(R*t)} * TN \tag{2}$$

where tissue $^{15}N$ atom% excess is the final $^{15}N$ atom% minus the initial $^{15}N$ atom%, $R$ is the mean $^{15}N$ atom% enrichment in the $NH_4^+$ pool, $t$ is the duration of the incubation, and $TN$ is the amount of nitrogen in the bull kelp tissues in μmoles.

All statistical tests, including ANOVA, ANCOVA, Tukey's Honestly Significant Difference, and t-tests, were performed in R (version 4.2.0), and figures were created in R and Microsoft Paint 3D.

## Ammonification inferred from metagenomes

Published metagenomes from the microbial community on Tatoosh and Squaxin bull kelp blades [21] allowed us to compare the potential for ammonification among the microbes on bull kelp at each locale by quantifying enzymes that hydrolyze carbon-nitrogen bonds within metagenome assembled genomes (MAGs). We used the International Union of Biochemistry and Molecular Biology designation (https://iubmb.qmul.ac.uk/enzyme/), and searched for enzymes with the nomenclature 'EC 1.4', which act on the CH-$NH_2$ group of donors, 'EC 3.5', which act on carbon-nitrogen bonds other than peptide bonds, and 'EC 4.3', carbon-nitrogen lyases within all 51 MAGs determined from *Nereocystis* blade surface samples collected at Squaxin (n = 13 MAGs) and Tatoosh (n = 38 MAGs) *Nereocystis* surfaces in July 2019 [20]. We only analyzed 2019 swab samples where host tissue contamination was minimized. We used the MAG database reported in Weigel et al. [21], which is available on the Figshare repository (https://figshare.com/s/84c036dc253a5dd1b1b9) and also archived at NCBI (PRJNA783443).

## Ethics statement

Samples used in this study were collected with the permission of the Makah Tribal Nation and the Washington Department of Natural Resources.

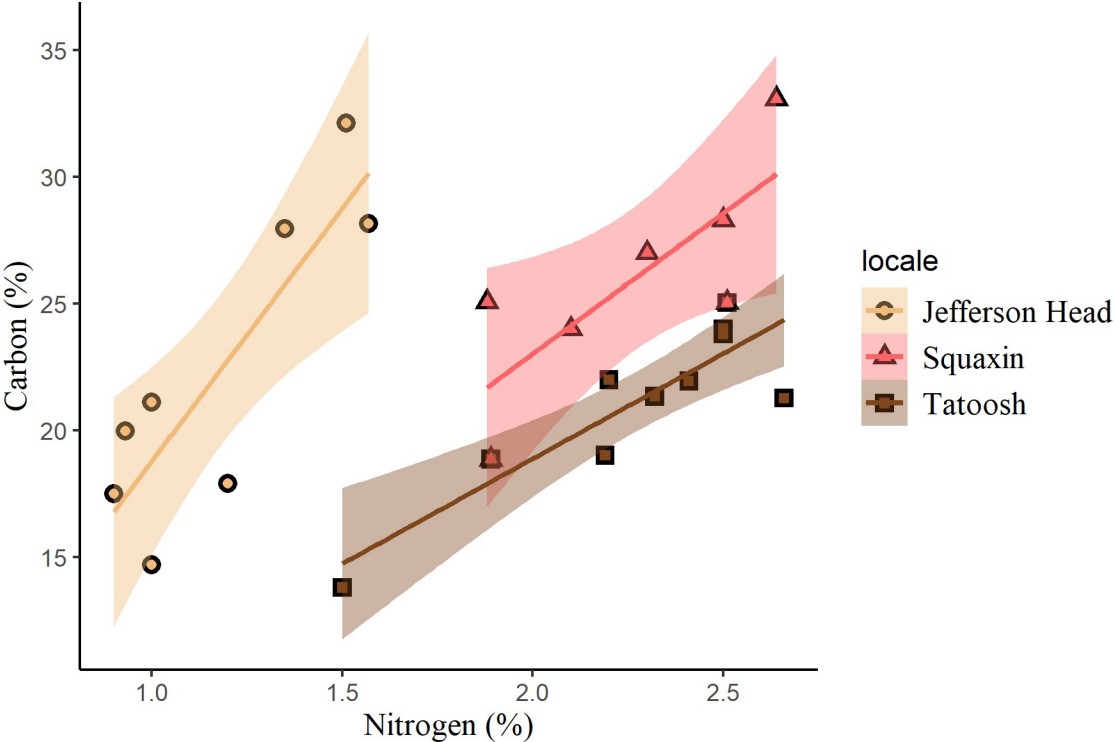

**Fig 2. Percent carbon vs. percent nitrogen.** The % carbon to % nitrogen content for *Nereocystis* blade tissue from three locales (n = 12). Bands represent 95% confidence intervals. The relationship between carbon and nitrogen differed significantly among sites for *Nereocystis* (ANCOVA, p = 0.011, $F_{2,21}$ = 5.616).

## Results

### The carbon and nitrogen content of hosts varied by site

Tissue carbon and nitrogen content differed among locales (Fig 2). Tissue nitrogen in *Nereocystis* was greatest in areas with the highest nitrogen concentration in the seawater. The seawater DIN concentration (sum of $NO_3$, $NO_2$, and $NH_4$ concentrations) was greatest at Tatoosh (Table 1), which had a mean concentration of 26.8 μM. Consequently, Tatoosh *Nereocystis* had the highest nitrogen tissue content, and the lowest C:N ratio. The seawater DIN concentration

**Table 1. Nereocystis incubations.**

| Location | Average PAR (μmol*m⁻²*s⁻¹) | Seawater Temperature (˚C) | Tissue C:N | Nutrient Concentration (μM) | | | | | Ammonification (nmol N L⁻¹ hr⁻¹) | | Kelp N uptake (μmol N hr⁻¹ g DW⁻¹) |
|---|---|---|---|---|---|---|---|---|---|---|---|
| | | | | $PO_4^{3-}$ | SI $(OH)_4$ | $NO_3^-$ | $NO_2^-$ | $NH_4^+$ | Seawater | Seawater w/ Kelp | |
| Jefferson Head | 946.3 ± 98.9 | 18.6 | 18.8 ± 1.0 | 0.47 | 3.75 | 0.74 | 0.00 | 0.57 | 0.02 ± 1x10⁻³ | 0.07 ± 0.01 | 3.0 ± 0.52 |
| Tatoosh | 1186 ± 190 | 11.7 | 9.34 ± 0.2 | 2.24 | 28.29 | 23.39 | 1.11 | 2.31 | 0.41 ± 2x10⁻³ | 0.41 ± 0.02 | 4.3 ± 0.27 |
| Squaxin | 959.4 ± 128 | 17.5 | 12.1 ± 0.3 | 1.60 | 39.56 | 3.13 | 0.15 | 2.08 | 0.19 ± 0.07 | 0.73 ± 0.07 | 5.2 ± 0.43 |

Incubations of *Nereocystis* from the coast of Washington State. Location names represent distinct bull kelp beds. We present data collected on tissue C:N ratio, seawater nutrient concentrations, ammonification rates, macrophyte N uptake rates, and conditions during the incubation. S1 Fig provides an illustration of the experimental chambers.

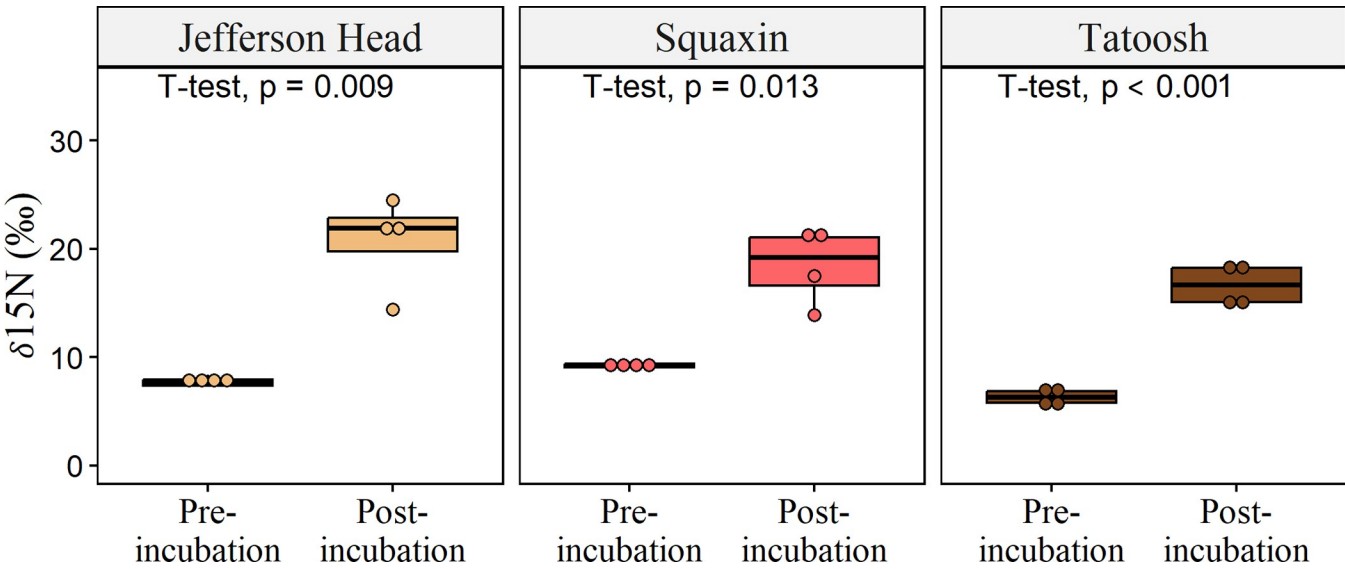

**Fig 3. Tissue 15N enrichment.** Pre-incubation and post-incubation $\delta^{15}$N enrichment levels in *Nereocystis* tissue. Pre- and post- incubation means were compared with t-tests, which are shown at the top of each panel (n = 4 for each).

was lower at the Puget Sound sites, with 5.21 μM at Squaxin, and 1.31 μM at Jefferson Head on the days of the incubations, correlating with higher C:N ratios. The slope of carbon to nitrogen differed among the three locales (Fig 2A, ANCOVA, p = 0.011, $F_{2,21}$ = 5.616), and nitrogen is greatest per unit carbon in Tatoosh kelp blades.

The nutrient concentrations of the seawater in incubation chambers with macrophytes decreased over the three-hour period while tissues increased in $\delta^{15}$N (Fig 3). $PO_4^{3-}$ decreased by 0.17 ± 0.06 μM/hr on average, $Si(OH)_4$ decreased by 2.60 ± 0.88 μM/hr, $NO_3^-$ decreased by 2.52 ± 0.78 μM/hr, $NO_2^-$ decreased by 0.11 ± 0.04 μM/hr, and $NH_4^+$ decreased by 0.13 ± 0.06 μM/hr. Prior to enrichment, the natural abundance of $\delta^{15}$N in sample tissue averaged 7.6‰ ± 0.2 in kelp. Following incubation, $\delta^{15}$N signatures increased significantly to 18.6‰ ± 1.0 (Fig 3).

### Ammonification and host nitrogen uptake

We used $\delta^{15}NH_4$ values in chamber seawater to calculate ammonification using Eq (1). Initial $\delta^{15}NH_4$ values of seawater in incubation chambers averaged -3.39‰ ± 0.32 before incubation. Following the incubations, $\delta^{15}NH_4$ values were enriched to 85.6‰ ± 21 in seawater only chambers and 129‰ ± 19 in kelp chambers, indicating that ammonification occurred. Ammonification associated with bull kelp differed significantly between each locale; despite lower tissue nitrogen, Squaxin bull kelp had the highest ammonification rate (Fig 4A). Nitrogen uptake by bull kelp, quantified with Eq (2), was lowest at Jefferson Head (Fig 4B). One sample from Squaxin appeared to have been contaminated based on greatly elevated nutrient measurements compared to other samples and was removed, resulting in n = 5 chambers for Squaxin.

Ammonification rates in control chambers with only seawater varied among sites and dates, with Tatoosh showing the greatest ammonification rates (Table 1). Macrophyte chambers displayed increased enrichment relative to control chambers for all bull kelp (Fig 5). The relationship between ammonium uptake and ammonification was positive for *Nereocystis* (S2 Fig).

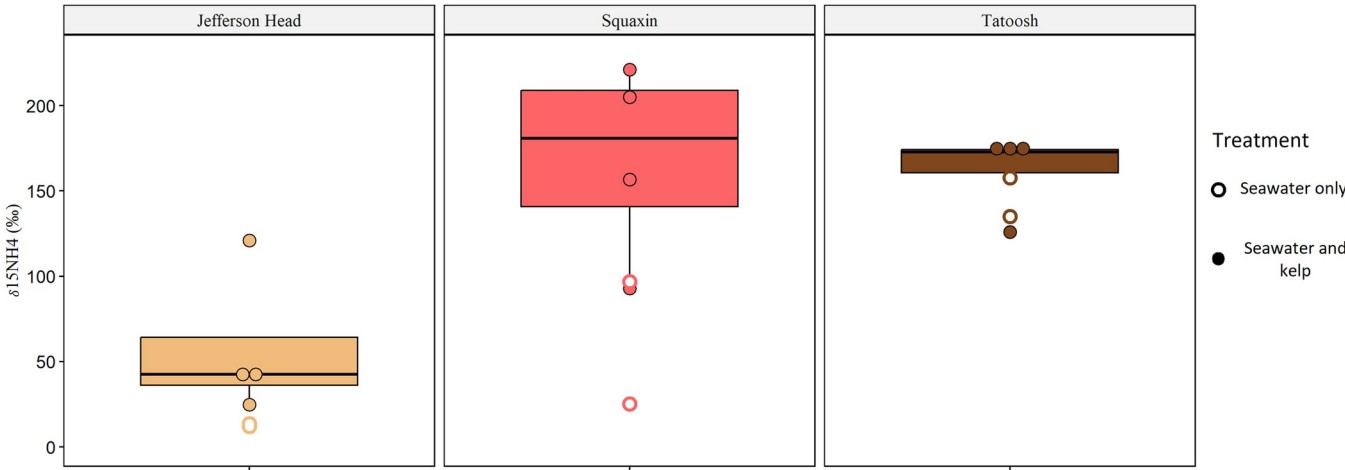

**Fig 4. Ammonification and nitrogen uptake.** Ammonification and nitrogen uptake rates for *Nereocystis*. Differences between locations for levels of *Nereocystis* ammonification were significant (p < 0.001), and nitrogen uptake was also significant (p = 0.020). There are n = 4 samples for most categories, though Squaxin Island *Nereocystis* had n = 3 samples because one was determined to be an outlier.

**Fig 5. Seawater 15N enrichment.** $\delta^{15}NH_4$ enrichment of the seawater in incubation chambers following the experiment. Open circles denote seawater-only control chambers (n = 2 per incubation) while filled circles indicate chambers with macrophytes (n = 4 per incubation).

## Metabolic function inferred from metagenomes

The number of genes detected in the Tatoosh bull kelp microbiome was higher than Squaxin by a factor of 2.6, with 80,633 genes at Tatoosh compared with 30,822 genes at Squaxin (Table 2). The overall number of ammonification genes, those with enzyme prefixes of EC1.4, EC3.5 and EC4.3, was lower at Squaxin, though it was proportional to the total functional gene discovery at 1.16% (Table 2). Every MAG at both Tatoosh and Squaxin contained ammonification genes in one of the 3 categories (Table 2), further indicating the likely high prevalence of this function.

Microbial gene detection in both Squaxin Island and Tatoosh Island metagenome assembled genomes (MAGs) reported in Weigel et al. [19]. Counts of KOfam genes and their overall percent representation are given. KOfam genes were counted as present at a cutoff of E-100.

# Discussion

## Bacterial contribution of DIN to seagrass and bull kelp hosts

Enrichment of $\delta^{15}NH_4$ occurred both in chambers with macrophytes and chambers without, indicating microbial transformation of DON into DIN by both water column microbes and macrophyte-associated microbes. We observed $\delta^{15}N$ uptake from our amino acid additions in all kelp samples (Fig 3), suggesting that ammonifying microbes contributed nitrogen to macrophytes through remineralization of DON. We further observed increased enrichment compared to control chambers in all chambers (Fig 5), suggesting that epiphytic microbes may also remineralize DON and further increase nitrogen availability for their hosts. This interpretation is supported by previous imaging of the microscale transfer of ammonium by nanoscale secondary ion mass spectrometry (NanoSIMS), which showed nitrogen isotopes passing from seawater into microbes, and from microbes into the surface layers of seagrass blades [16]. This phenomenon may be widespread in macrophyte-microbiome systems. For instance, the functional capacity for ammonification was enriched in the giant kelp *Macrocystis* compared with surrounding seawater [36], as well as in cultures with the green alga *Ulva* [37], and ammonification was enhanced over tropical seagrass systems [12].We note that in our study, and in many other host-microbe systems, elemental imaging would facilitate efforts to definitively trace nutrient exchanges between hosts and their microbiome.

Macrophyte nitrogen uptake was consistently 4–5 orders of magnitude greater than ammonification in the incubation chambers, ranging from 3.0–5.2 µmol N g DW$^{-1}$ hr$^{-1}$ (Table 1). Ammonium uptake rates by macrophytes were also large compared to ammonium pools in the seawater. However, despite the relatively low ammonification rates relative to nitrogen uptake, complete depletion of inorganic nitrogen did not occur in the chambers. One explanation is that algal uptake estimates were based on the enrichment in ammonium (Eq (2) above), while ammonification rates relied on our estimates of enrichment of amino acids. Amino acid concentrations are poorly estimated in these locales, and we assumed 1 µM based on previously published studies [29,30]. We note that if amino acid concentrations were greater than

**Table 2. Microbial gene detection.**

| Genes | Squaxin Island | Tatoosh Island |
|---|---|---|
| EC1.4 | 90 (0.29%) | 265 (0.33%) |
| EC3.5 | 184 (0.60%) | 461 (0.57%) |
| EC4.3 | 82 (0.27%) | 210 (0.26%) |
| All ammonification genes | 356 (1.16%) | 936 (1.16%) |
| Total gene tally | 30,822 | 80,663 |

1 μM, we would have overestimated enrichment ($R_{source}$) and underestimated ammonification in Eq (1). DON estimates in the coastal northeast Pacific and at these sites can exceed 10 μM [38]. If we assume amino acid concentrations as high as 10 μM, then our estimates of ammonification would increase by an order of magnitude. Secondly, if spatially localized processes were important, such as immediate microbial production followed by host use of ammonium at the macrophyte surface biofilm, we would also underestimate ammonification in the water column. A third explanation may be the rapid uptake and production of inorganic nitrogen by the multitude of metabolisms in intertidal systems, which lead to quick turnover in nitrogen pools in seawater [10,20,34,39]. Because we only took measurements of seawater nutrients at the initiation and end of the incubations, the processing and recycling of nitrogen may have been underestimated.

One important alternative to consider is that kelp may directly take up DON, with enriched amino acids entering the host without microbial intermediaries. Phytoplankton have been shown to have amino acid oxidases on the cell surface that produce ammonium from amino acids [9]. Previous studies have also observed amino acid oxidases and amino acid uptake in seaweeds. For instance, amino acid uptake was demonstrated during rhizoid development in *Fucus* with enriched carbon in amino acids and enriched sulfur in methionine [40]. Other observations of amino acid oxidases have been attributed to the host seaweed [41–45]. Further, there are several studies that also suggest uptake of DON by other seaweeds and seagrasses [14,46–49]. In all of these studies, however, the role of bacterial metabolisms in nitrogen acquisition remains unclear because it is not reported whether axenic conditions were strictly maintained during incubations. Regardless, some studies do provide strong inference for direct uptake of amino acids. For instance, studies of *Fucus* embryos have shown evidence of normal development of *Fucus* embryos or zygotes when the amino acid methionine is the only source of sulfur [40–42], and images show that embryos seem to be bacteria-free [42]. Furthermore, the use of stable isotope tracing indicated that urea can be taken up by the giant kelp *Macrocystis* [15], though not by the kelp *Ecklonia* [50]. Still, Tarquinio et al.'s [16] work with the seagrass *Posidonia*, where elemental imaging of isotopes of nitrogen was used in the presence and absence of bacteria, provides strong inference that microbial ammonification produced ammonium and that it was $^{15}NH_4$ that was taken up by the host. Furthermore, metagenomic sampling demonstrated that genes with the capacity to cleave C-N bonds in amino acids were common in bacteria residing on *Nereocystis*. The lack of clear axenic conditions in previous studies and evidence from elemental imaging support the importance of microbial metabolisms in nitrogen uptake by *Nereocystis*. However, the possible diversity of amino acid metabolism routes suggested in the above referenced studies cannot be discounted, and needs further research efforts with imaging and isotope incubation, in combination with analysis of bacterial and seaweed genomes.

## Comparing bull kelp from different locales

If microbes play an important role in nitrogen acquisition for macrophytes, then the presence or absence of these microbes may have implications for the fitness of the host. Bull kelp samples collected from Tatoosh, Jefferson Head, and Squaxin differed in fitness and microbial abundance. Tatoosh Island bull kelp are exposed to the open ocean and exhibit the greatest tissue nitrogen content and more diverse microbial communities [22,51]. Squaxin Island kelp represent the southernmost population in Puget Sound. They have been in sharp decline [26], have reduced microbial diversity [22], sparse microbial communities on their surfaces [25], and lower nitrogen content (Fig 2). The Jefferson Head kelp bed is a restored feature where kelp grows on line substrates, and we have no information on microbial abundances there, though we note that nitrogen content in Jefferson Head kelp is extremely low.

The greater health and nitrogen content of the Tatoosh Island kelp led us to hypothesize that their microbiome would have greater ammonification rates and increased access to dissolved organic nitrogen. Our results show otherwise, and indicate that the microbiome on the smaller, more poorly growing Squaxin *Nereocystis* had the greatest ability to ammonify and process amino acids. Notably, Tatoosh Island had high ammonification rates in the water column, which may indicate that ammonification is not as restricted to host-association as it is at the other sites. Thus, Tatoosh Island, with the highest concentration of dissolved inorganic nitrogen in the surrounding waters (Table 1), and the greatest tissue nitrogen levels (Fig 2), had the lowest rates of ammonification by *Nereocystis*-associated microbes. Our results suggest that the relatively nitrogen-poor Squaxin kelp samples opportunistically took up nitrogen when it became available, a result consistent with Weigel and Pfister [28].

## Differences in seawater between sites

The composition of epiphytic communities on macrophytes may vary depending on the nutrient resources available to them [52]. Seawater nutrient conditions differed significantly between the three sampling locations. Tatoosh Island, which lies on the outer coast of Washington State, is supplied from deep areas offshore by upwelling, which occurs along much of the Pacific Northwestern coast [53]. Water from deep areas is cold and nutrient rich [54], likely supporting the robust local bull kelp and seagrass populations. Furthermore, inorganic sources of nitrogen, especially nitrate, are much higher in concentration at Tatoosh Island than locales in Puget Sound (Table 1) [55]. However, we currently know little about the concentration of DON components across these locales, the microbial taxa that can ammonify, and the ability of nitrogen-poor host species to select for different microbiomes. Our use of unfiltered seawater in our assays suggested that water column microbes in addition to host-associated microbes are quantitatively important here and may influence selection at the host-microbe interface. The noticeable increase in $\delta^{15}NH_4$ enrichment observed even in chambers that did not contain macrophytes (Fig 5) indicates that some ammonifying microbes may not always be in intimate association with the host. Because this metabolism could also benefit microbes, ammonifying microbes might be both free-living and host-associated.

## Metabolic patterns indicated by metagenomes

Metabolic information from genomics indicated the presence of approximately equivalent functional capacity across metagenomes from Squaxin and Tatoosh. Even though microbe density and diversity were much greater at Tatoosh [21,23], the proportional capacity for ammonification based on the presence of genes was equivalent and may account for the unexpectedly high rates of ammonification in *Nereocystis* from Squaxin. Although we do not have metagenomic samples in July 2021, when we did the experiments, repeated metagenomic sampling at Tatoosh, Weigel et al. 2022, [21] suggests similar metagenomes across years. There was no metagenomic information to test this for the *Nereocystis* from Jefferson Head.

Ammonification genes, those that cleave C-N bonds in amino acids and produce ammonium, were common in *Nereocystis* metagenomes analyzed here [6]. The giant kelp *Macrocystis* in the southern region of the California Current system also showed genes capable of ammonification in the surface microbiome [36].

Tatoosh Island bull kelp, which have been persistent and non-declining compared with the diminishing Squaxin Island population [26,27], exhibited microbial genes that were enriched with modules that might increase ammonification, including histidine regulation and proline metabolism. There is lower abundance and diversity of host-associated microbes at Squaxin Island [21,22,25], and ammonification genes are fewer. Yet, there was no indication from

amino acid incubations that the functional capacity of Squaxin to use DON was compromised compared to ammonification measures for healthy Tatoosh populations. In sum, our study suggests that DON, a significant component of the free nitrogen in coastal seawater, may be accessible to seaweed via the activities of associated microbes and points to a need to better understand how the microbiome of primary producers contribute to their fitness.

## Supporting information

**S1 Fig. Experimental chamber schematic.** A schematic of the hypothesized movement of enriched $^{15}$N in experimental chambers containing *Nereocystis* when added as $^{15}$N amino acids. Red arrows denote those we have quantified, including the microbially-mediated transfer of $^{15}$N from amino acids to ammonium via ammonification, a process that could have been done by host-associated or free-living microbes. Chambers without host kelp quantified water column ammonification only. The $^{15}$N measured in hosts was assumed to come from this ammonification. The dashed line shows bacterial uptake of ammonium that might have occurred but was not quantified.
(TIF)

**S2 Fig. Nitrogen uptake vs. ammonification.** Nitrogen uptake rate (in μmol) vs. ammonification rate (in nmol) for *Nereocystis* sampling locations. Linear regressions indicated a significant positive association between ammonification and nitrogen Uptake in bull kelp (p = 0.024, $r^2$ = 0.389). The bands around the regression line represents the 95% confidence interval.
(TIF)

## Acknowledgments

We thank the Makah Tribal Nation for access to Tatoosh Island. We thank H. Berry of the WDNR and G. McKenna, H. Hayford, B. Allen, and J. Davis of the Puget Sound Restoration Fund for kelp collections and boat access to Puget Sound sites. Assistance in the field from A. Wootton, J. T. Wootton, E. Shaw, and T. Fonseca was greatly appreciated. A. Masterson at Northwestern provided expertise in tissue isotope samples, J. Herszage at UC Davis in seawater isotopes, and A. Morello at University of Washington in seawater nutrient analyses. S. Kuehn, J. Waldbauer, R. Weed, B. Weigel, D. Claar, E. Stanfield, and two anonymous reviewers provided helpful comments on a previous version of the manuscript; S. Weinstein helped with S1 Fig.

## Author Contributions

**Conceptualization:** Alex Hochroth, Catherine A. Pfister.

**Data curation:** Alex Hochroth, Catherine A. Pfister.

**Formal analysis:** Alex Hochroth, Catherine A. Pfister.

**Funding acquisition:** Catherine A. Pfister.

**Investigation:** Alex Hochroth.

**Methodology:** Alex Hochroth, Catherine A. Pfister.

**Project administration:** Catherine A. Pfister.

**Resources:** Catherine A. Pfister.

**Supervision:** Catherine A. Pfister.

**Validation:** Catherine A. Pfister.

**Visualization:** Alex Hochroth.

**Writing – original draft:** Alex Hochroth.

**Writing – review & editing:** Alex Hochroth, Catherine A. Pfister.

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
