## [Decision Letter · Decision Letter 0]

25 Aug 2023

PONE-D-23-16300Ammonification by kelp associated microbes increases ammonium availabilityPLOS ONE

Dear Dr. Hochroth,

Thank you for submitting your manuscript to PLOS ONE. After careful consideration, we feel that it has merit but does not fully meet PLOS ONE’s publication criteria as it currently stands. Therefore, we invite you to submit a revised version of the manuscript that addresses the points raised during the review process.

Especially, please make sure to expand the method section with respect to amino acid concentrations within your treatments and consider to give more information on microbial composition of kelp-associated microbes. Also, please consider to incorporate older studies mentioned by the reviewers in your discussion and carefully edit the ms.

We look forward to receiving your revised manuscript.

Kind regards,

Frank Melzner

Academic Editor

PLOS ONE

Journal Requirements:

4.  We note that Figure 1 in your submission contain map images which may be copyrighted. All PLOS content is published under the Creative Commons Attribution License (CC BY 4.0), which means that the manuscript, images, and Supporting Information files will be freely available online, and any third party is permitted to access, download, copy, distribute, and use these materials in any way, even commercially, with proper attribution. For these reasons, we cannot publish previously copyrighted maps or satellite images created using proprietary data, such as Google software (Google Maps, Street View, and Earth). For more information, see our copyright guidelines: http://journals.plos.org/plosone/s/licenses-and-copyright.

Reviewers' comments:

Reviewer's Responses to Questions

**Comments to the Author**

1. Is the manuscript technically sound, and do the data support the conclusions?

Reviewer #1: Partly

Reviewer #2: Partly

2. Has the statistical analysis been performed appropriately and rigorously? 

Reviewer #1: Yes

Reviewer #2: Yes

3. Have the authors made all data underlying the findings in their manuscript fully available?

Reviewer #1: Yes

Reviewer #2: Yes

4. Is the manuscript presented in an intelligible fashion and written in standard English?

Reviewer #1: Yes

Reviewer #2: Yes

5. Review Comments to the Author

Reviewer #1: Dear authors!

Your manuscript describes a work that aimed to investigate the role of ammonifying bacteria associated with Nereocystis for nitrogen availability to the host alga. You have combined two different methodological approaches. On the one hand, you determined the uptake rate of 15N into the algae and the release of 15NH4 into the medium in incubation experiments with 15N-labeled amino acids and you observed, that nitrogen from amino acids accumulates within 3 h both in kelp tissue and seawater ammonium. On the other hand, you determined the relative abundance of bacterial genes that can contribute to ammonification in surface swabs of free-living algae and you observed presence such microbiota, but no clear correlation between ammonification and tissue nitrogen. You conclude that microbial ammonification may contribute to the availability of ammonium to kelps. In general, bacterial degradation of amino acids would be expected to contribute to the availability of ammonium to algae such as Nereocystis, but there is little actual evidence to date. Your study supports this expectation and is therefore relevant and should be published. However, I see two problems in your study or its presentation.

(1) The first problem is the fact that you do not provide clear information about the initial concentration of amino acids in your experiments. The experiments were performed with natural sea water, whose amino acid content you could not determine. You write that you assume a concentration of about 1 µmol, because this is the concentration that is mostly mentioned in the literature (which is correct). You then added a mixture of 15N-labeled amino acids to this seawater and this is where the problems start. In l. 120 you write you added 200 µL of a 0.25M stock solution into an incubation chamber containing 2.6 L sea water (l. 110). If I got this right this would mean the initial concentration of labeled amino acids was 19.23 µmol. Yet, in l. 171 you write your initial concentration in the chamber was 0.25M (!) and in line 128 you write the total initial amino acid concentration (labeled and unlabeled) may have been as high as 524.6 µM. These three informations obviously do not match. In addition I was unable to follow the argument you make to explain the last of the three values: You write your addition was 525.43 o/oo of the supposed natural background concentration of 1 µM, together this would be 1.52543 µM rather than 524.6 µM? Please clarify this. Ignoring the natural background if the initial added concentration was 250 mM would be no issue. Ignoring it if the addition was 525.43 nM would be one.

Also, you provide no information on the composition of your labeled amino acids, but this information is important, as not all amino acids are metabolized in the same way.

(2) In the discussion you mention briefly the possibility that Nereocystis itself could contribute to ammonification via amino acid oxidase, but then you dismiss this possibility with the argument that you are simply not aware of any reports of such mechanisms in seaweeds. However, they exist. Please see

Fujisawa et al 1982. Occurence of L-amino acid oxidase in the marine red alga Gymnogongrus flabelliformis. Bulletin of the Japanese Society for Scientific Fisheries 48, 97–103.

Ito et al. 1987. Purification and some properties of L-amino acid oxidase from Amphiroa crassissima Yendo. Hydrobiologia 151/152, 563–569.

Weinberger et al. 2005. Apoplastic oxidation of L-asparagine is involved in the control of the green algal endophyte Acrochaete

operculata Correa & Nielsen by the red seaweed Chondrus crispus Stackhouse. J. Exp. Bot. 56:1317-1326.

In particular the last study seems relevant in your context because the authors demonstrated not only the presence of a cell-wall-located amino acid oxidase that released NH4 from amino acids, but they also observed evidence of an algal uptake of 14C-labelled amino acids within very short time. Obviously such uptake could be another alternative explanation for increasing 15N-tissue concentrations in your study that should not be simply ignored.

If Nereocystis would be capable of amino acid oxidation then this might be tested via measurement of H2O2 medium concentrations after addition of amino acids to the medium. Perhaps something to consider!

Also:

l 388 what is meant with the comment in brackets?

Fig. 4: ANOVA probably in capitals?

Reviewer #2: The extent to which ammonification by kelp microbes increases ammonium availability is an important research question, and so is coastal amino acid concentration.

Line 46-47: I think this information from Hurd and the three refs below may have some information of interest to authors:

Hurd et al. (2nd ed Seaweed Ecology & Physiology, 2014) write (p. 258, 2nd Paragraph, “ There has been only one definitive study on the uptake kinetics of amino acids. Schmitz & Riffarth ..examined…17 amino acids (uptake) by Hincksia mitchelliae…(they) suggested that exogenous amino acids contribute <5% of the N demand by this alga”. But the refs below might suggest more uptake by host proper of amino acids that support development?

Contrast this with:

1. Hogsett & Quatrano. 1978. J. Cell Biol. 78, 866-873.

2. Crayton et al. 1974. Develop. Biol. 39, 164-167.

3. Brawley & Quatrano. 1979. Develop. Biol. 73, 193-205.

Line 93. Please give source of knowledge of previous natural bed at Jefferson Head.

Line 100-101: Why were the numbers of chambers between seawater only and seawater plus kelp not balanced? What effect has this had on your results?

Hasty preparation of ms (examples; there are more in ms so proof carefully!):

Line 122 ---missing comma in compound sentence

Line 123---µM is the correct unit (Not uM): this error is common in figure axes, legends and in text: correct throughout ms.

Line 264: italicize Nereocystis

Line 388: edit by senior author not removed

REFERENCE LIST: References often lack italics for species names OR are EITHER in caps or not etc. PLEASE proof and reformat references, uniformly, to PLoS preferences.

Lines 121-122: The ms would be much stronger if aqueous amino acid concentration had been measured at each site; it may be significantly different between Tatoosh and other sites, especially Jefferson Point?

Lines 129-133: Give seawater temperature at each site during incubations AND PAR levels (µmol m2/s): these all affect your experiments in quantitative ways!

Line 141: What was syringe pore-size? 0.2 µm?

Line 182: µmoles ?

Line 197: CLASI-FISH suggests that swabs are inadequate to sample the microbiome?

LINES 210-213: This sentence is poorly written (and not accurate?).

Line 203: What was DON variation by site?

LINE 222: Add temperature and irradiance levels.

LINE 257: Easier to be confident in “outlier” elimination if sample sizes were higher.

LINES 271-285: I find it unsatisfying to not have a better description of the composition of the microbiomes from the 3 sites. Perhaps this information is in an earlier (related), published paper? If so, please ensure it is transparent here; if not, please consider adding at a minimum a bar graph and full list of taxonomic assignments for ASVs.

LINE 272-276: IT IS UNCLEAR TO ME WHETHER THE NUMBER OF BACTERIA DIFFERS BETWEEN SITES OR THE NUMBER OF GENES/ASV.

Line 311: Nereocystis is not a plant. It is a member of the Stramenopila supergroup.

Lines 328-335: See comments at beginning of this review, and read Hurd et al. (2nd ed) Chapter6, especially 6.5.1 (Nitrogen) and revise.

LINES 348-358: Differences in DON (not measured) across sites, and selection for different microbiomes, could contribute significantly, too.

LINES 368-371: Yes!

6. PLOS authors have the option to publish the peer review history of their article (what does this mean?). If published, this will include your full peer review and any attached files.

Reviewer #1: No

Reviewer #2: No

---

## [Author Response · Author response to Decision Letter 0]

9 Oct 2023

Dear Dr. Melzner:

We are grateful to receive two expert reviews and thank both reviewers for insightful comments on our manuscript, PONE-D-23-16300, “Ammonification by kelp associated microbes increases ammonium availability”. We have revised the manuscript based on these comments and are pleased to resubmit this improved version for your inspection. We detail our revisions below.

Editor Melzner:

Especially, please make sure to expand the method section with respect to amino acid concentrations within your treatments and consider to give more information on microbial composition of kelp-associated microbes. Also, please consider to incorporate older studies mentioned by the reviewers in your discussion and carefully edit the ms.

Our revisions are detailed below and address the overall concerns by the Editor.

Our Ethics Statement is now included at the end of the Methods (L207).

We have added the location of the minimal data set to the Data Availability Statement (L421). Data can be found in the Figshare repository at the following url: https://figshare.com/projects/Plos_One_Nereocystis_Ammonification/180562

There are no legal restrictions to making the data public.

4. We note that Figure 1 in your submission contain map images which may be copyrighted. All PLOS content is published under the Creative Commons Attribution License (CC BY 4.0), which means that the manuscript, images, and Supporting Information files will be freely available online, and any third party is permitted to access, download, copy, distribute, and use these materials in any way, even commercially, with proper attribution. For these reasons, we cannot publish previously copyrighted maps or satellite images created using proprietary data, such as Google software (Google Maps, Street View, and Earth). For more information, see our copyright guidelines: http://journals.plos.org/plosone/s/licenses-and-copyright.

We have updated Figure 1 to use map data exclusively from Natural Earth, which is public domain. The map was created with QGIS 3.32.3, an open source mapping software. There should be no copyright concerns remaining. The data source has also been added to the caption of Figure 1 (L105).

Reviewer #1: Dear authors!

Your manuscript describes a work that aimed to investigate the role of ammonifying bacteria associated with Nereocystis for nitrogen availability to the host alga. You have combined two different methodological approaches. On the one hand, you determined the uptake rate of 15N into the algae and the release of 15NH4 into the medium in incubation experiments with 15N-labeled amino acids and you observed, that nitrogen from amino acids accumulates within 3 h both in kelp tissue and seawater ammonium. On the other hand, you determined the relative abundance of bacterial genes that can contribute to ammonification in surface swabs of free-living algae and you observed presence such microbiota, but no clear correlation between ammonification and tissue nitrogen. You conclude that microbial ammonification may contribute to the availability of ammonium to kelps. In general, bacterial degradation of amino acids would be expected to contribute to the availability of ammonium to algae such as Nereocystis, but there is little actual evidence to date. Your study supports this expectation and is therefore relevant and should be published. However, I see two problems in your study or its presentation.

(1) The first problem is the fact that you do not provide clear information about the initial concentration of amino acids in your experiments. The experiments were performed with natural sea water, whose amino acid content you could not determine. You write that you assume a concentration of about 1 µmol, because this is the concentration that is mostly mentioned in the literature (which is correct). You then added a mixture of 15N-labeled amino acids to this seawater and this is where the problems start. In l. 120 you write you added 200 µL of a 0.25M stock solution into an incubation chamber containing 2.6 L sea water (l. 110). If I got this right this would mean the initial concentration of labeled amino acids was 19.23 µmol. Yet, in l. 171 you write your initial concentration in the chamber was 0.25M (!) and in line 128 you write the total initial amino acid concentration (labeled and unlabeled) may have been as high as 524.6 µM. These three informations obviously do not match. In addition I was unable to follow the argument you make to explain the last of the three values: You write your addition was 525.43 o/oo of the supposed natural background concentration of 1 µM, together this would be 1.52543 µM rather than 524.6 µM? Please clarify this. Ignoring the natural background if the initial added concentration was 250 mM would be no issue. Ignoring it if the addition was 525.43 nM would be one.

Also, you provide no information on the composition of your labeled amino acids, but this information is important, as not all amino acids are metabolized in the same way.

We regret our confusing presentation on the concentration of amino acids and thank the reviewer for catching this. The amount added to the 2.6L chambers was 200µL of a 0.025M (we mistakenly wrote 0.25M in the original ms), so the concentration of 15N amino acids was 1.92 µM. We have corrected the text (L120), but this did not affect our original uptake calculations. 

The amino acids that are present are important, although we do not know amino acid specific metabolisms. The mix that we used (cited in the paper as Cambridge Isotope Labs, product# NLM-2161, Lot# PR-24163) is a mix of 16 amino acids, with leucine, glutamic acid and alanine contributing the most and making up the greatest percentage 37% of the total. The amino acids valine, phenylalanine, isoleucine, aspartic acid, glycine, threonine, proline, tyrosine, arginine, lysine, serine, methionine, histadine all composed between 2 and 9%. We now include that information in the Methods (L124-128), though we recognize that how these amino acids were individually metabolized remains unknown.

(2) In the discussion you mention briefly the possibility that Nereocystis itself could contribute to ammonification via amino acid oxidase, but then you dismiss this possibility with the argument that you are simply not aware of any reports of such mechanisms in seaweeds. However, they exist. Please see

Fujisawa et al 1982. Occurence of L-amino acid oxidase in the marine red alga Gymnogongrus flabelliformis. Bulletin of the Japanese Society for Scientific Fisheries 48, 97–103.

Ito et al. 1987. Purification and some properties of L-amino acid oxidase from Amphiroa crassissima Yendo. Hydrobiologia 151/152, 563–569.

Weinberger et al. 2005. Apoplastic oxidation of L-asparagine is involved in the control of the green algal endophyte Acrochaete

operculata Correa & Nielsen by the red seaweed Chondrus crispus Stackhouse. J. Exp. Bot. 56:1317-1326.

In particular the last study seems relevant in your context because the authors demonstrated not only the presence of a cell-wall-located amino acid oxidase that released NH4 from amino acids, but they also observed evidence of an algal uptake of 14C-labelled amino acids within very short time. Obviously such uptake could be another alternative explanation for increasing 15N-tissue concentrations in your study that should not be simply ignored.

We note that these are interesting and relevant studies and have incorporated them into our manuscript. We agree that we cannot dispute the presence of amino acid oxidases in the host bull kelp and that their presence would increase d15N in the host or microbiome. We expanded our Discussion to incorporate this literature and the suggestion of amino acid uptake and amino acid oxidases (Discussion L331-333). However, the Fuijsawa et al. paper, as well as the Ito et al. paper - though interesting - do not consider nor provide proof that microbial enzymes were important. We emphasize that the strongest inference for host amino acid oxidases will continue to come from stable isotope analyses and detailed imaging in tandem with whole genome sequences of the host. We hope our reported results inspire more investigation in this area. 

If Nereocystis would be capable of amino acid oxidation then this might be tested via measurement of H2O2 medium concentrations after addition of amino acids to the medium. Perhaps something to consider!

Indeed, it would be interesting to know if bacteria elicited the same response in the host that the endophytic green alga did, as in Weinberger 2005. Antibiotics would also be an interesting future manipulation in this system.

Also:

l 388 what is meant with the comment in brackets?

Apologies. It was a missed citation.

Fig. 4: ANOVA probably in capitals?

Yes, ANOVA is now capitalized on Figure 4.

Reviewer #2: The extent to which ammonification by kelp microbes increases ammonium availability is an important research question, and so is coastal amino acid concentration.

agreed. 

Line 46-47: I think this information from Hurd and the three refs below may have some information of interest to authors:

Hurd et al. (2nd ed Seaweed Ecology & Physiology, 2014) write (p. 258, 2nd Paragraph, “ There has been only one definitive study on the uptake kinetics of amino acids. Schmitz & Riffarth ..examined…17 amino acids (uptake) by Hincksia mitchelliae…(they) suggested that exogenous amino acids contribute <5% of the N demand by this alga”. But the refs below might suggest more uptake by host proper of amino acids that support development?

The Schmitz and Riffarth paper, though interesting, did not specifically state they had axenic cultures nor did they assay or quantify bacteria, making it unclear whether the amino acid Leucine was taken up solely by the alga. The references below are more suggestive.

Contrast this with:

1. Hogsett & Quatrano. 1978. J. Cell Biol. 78, 866-873.

2. Crayton et al. 1974. Develop. Biol. 39, 164-167.

3. Brawley & Quatrano. 1979. Develop. Biol. 73, 193-205.

We thank Rev 2 for these highly relevant citations. As we state above, it is difficult to have strong inference for host uptake of amino acids or amino acid oxidases independent of associated microbes. However, we appreciate these developmental studies given their use of isotopes and microscopy and they do provide inference of amino acid uptake. We cite these on L333.

Line 93. Please give source of knowledge of previous natural bed at Jefferson Head.

The knowledge of a kelp bed at Jefferson Head is based on oral history from the Suquamish Tribe and is thus indigenous in origin. We cannot find a published account of this, but tribal elders recall bull kelp at this site and that was a dominant reason for the private group, Puget Sound Restoration Fund, situating the restoration there and the state of Washington approving it.

Line 100-101: Why were the numbers of chambers between seawater only and seawater plus kelp not balanced? What effect has this had on your results?

We had fewer controls because we know from previous work that seawater only is less dynamic in carbon and nitrogen fluctuations compared with chambers with kelp. We duplicated the chambers containing seawater only to estimate and subtract this change due only to seawater from what we quantified due to the presence of a kelp blade. The ‘unbalanced’ number of chambers was not used in a strict ANOVA design, for example. Instead, duplicate controls in the field allowed us to quantify and control for background seawater activity at each site.

Hasty preparation of ms (examples; there are more in ms so proof carefully!):

Line 122 ---missing comma in compound sentence

Corrected with comma inserted (L129). 

Line 123---µM is the correct unit (Not uM): this error is common in figure axes, legends and in text: correct throughout ms.

We understand, though sometimes it is less error prone to insert symbol fonts when ‘u’ is understood to be ‘micro’ in some journals. We will insert the more precise µM.

Line 264: italicize Nereocystis

Corrected.

Line 388: edit by senior author not removed

Corrected. 

REFERENCE LIST: References often lack italics for species names OR are EITHER in caps or not etc. PLEASE proof and reformat references, uniformly, to PLoS preferences.

Corrected. 

Lines 121-122: The ms would be much stronger if aqueous amino acid concentration had been measured at each site; it may be significantly different between Tatoosh and other sites, especially Jefferson Point?

Agreed, though we simply do not have this information and quantifying amino acids is a complex and expensive undertaking, given the numerous amino acids and their presence as both total and free amino acid forms. We hope that by highlighting the potential importance of amino acids, there is increased funding and motivation to measure them accurately.

Lines 129-133: Give seawater temperature at each site during incubations AND PAR levels (µmol m2/s): these all affect your experiments in quantitative ways!

We have added the PAR and temperature values at the day and time of the incubation to Table 1 (L228). We incubated in natural light and with ambient temperatures.

Line 141: What was syringe pore-size? 0.2 µm?

Whatman gf/f are 0.7 µm and that is what we used. This has been added to the Methods section at L148.

Line 182: µmoles?

Yes, corrected

Line 197: CLASI-FISH suggests that swabs are inadequate to sample the microbiome?

CLASI-FISH does suggest that there are microbes that are under the surface kelp cells, though it is still unclear if they are different genera and/or strains compared to those in the biofilm.

LINES 210-213: This sentence is poorly written (and not accurate?).

The sentence has been corrected to the following:

“The slope of carbon to nitrogen differed among the three locales (Figure 2a, ANCOVA, p=0.011, F2,21= 5.616), and nitrogen is greatest per unit carbon in Tatoosh kelp blades.” (L219-221)

Line 203: What was DON variation by site?

We regret that we do not have this information, nor what percentage of the DON is amino acids.

LINE 222: Add temperature and irradiance levels.

Temperature and average irradiance levels were added in Table 1 (L228).

LINE 257: Easier to be confident in “outlier” elimination if sample sizes were higher.

Agreed.

LINES 271-285: I find it unsatisfying to not have a better description of the composition of the microbiomes from the 3 sites. Perhaps this information is in an earlier (related), published paper? If so, please ensure it is transparent here; if not, please consider adding at a minimum a bar graph and full list of taxonomic assignments for ASVs.

At the time this study was done, we did not have funding to simultaneously quantify the microbes at these sites. Hence, we use published information from our previous analyses.

LINE 272-276: IT IS UNCLEAR TO ME WHETHER THE NUMBER OF BACTERIA DIFFERS BETWEEN SITES OR THE NUMBER OF GENES/ASV.

Bacteria were denser and more diverse at Tatoosh than Squaxin. However, the proportional representation of ammonifying genes was similar between sites.

Line 311: Nereocystis is not a plant. It is a member of the Stramenopila supergroup.

Indeed. Replaced with ‘algal’.

Lines 328-335: See comments at beginning of this review, and read Hurd et al. (2nd ed) Chapter6, especially 6.5.1 (Nitrogen) and revise.

We have revised this section of the Discussion, though we note that strong inference of direct amino acid uptake by macroalgae is rare. The Schmitz paper cited does not state whether the alga was in axenic culture, nor were bacteria assayed. Thus, it remains uncertain whether the bull kelp host has the ability to directly take up amino acids. We note this in the Discussion and include: “…more such studies with imaging and isotope incubation, in combination with analysis of seaweed genomes, are needed.” (L341-342).

LINES 348-358: Differences in DON (not measured) across sites, and selection for different microbiomes, could contribute significantly, too.

Agreed, and we note this with a revised statement on L377.

LINES 368-371: Yes!

Agreed.

---

## [Decision Letter · Decision Letter 1]

18 Dec 2023

Ammonification by kelp associated microbes increases ammonium availability

PONE-D-23-16300R1

Dear Dr. Hochroth,

We’re pleased to inform you that your manuscript has been judged scientifically suitable for publication and will be formally accepted for publication once it meets all outstanding technical requirements. Please make sure to incorporate the very helpful suggestions / explanations reviewer 2 supplied.

Kind regards,

Frank Melzner

Academic Editor

PLOS ONE

Additional Editor Comments (optional):

Reviewers' comments:

Reviewer's Responses to Questions

**Comments to the Author**

1. If the authors have adequately addressed your comments raised in a previous round of review and you feel that this manuscript is now acceptable for publication, you may indicate that here to bypass the “Comments to the Author” section, enter your conflict of interest statement in the “Confidential to Editor” section, and submit your "Accept" recommendation.

Reviewer #1: All comments have been addressed

Reviewer #2: (No Response)

2. Is the manuscript technically sound, and do the data support the conclusions?

Reviewer #1: Yes

Reviewer #2: Yes

3. Has the statistical analysis been performed appropriately and rigorously? 

Reviewer #1: Yes

Reviewer #2: Yes

4. Have the authors made all data underlying the findings in their manuscript fully available?

Reviewer #1: Yes

Reviewer #2: Yes

5. Is the manuscript presented in an intelligible fashion and written in standard English?

Reviewer #1: Yes

Reviewer #2: Yes

6. Review Comments to the Author

Reviewer #1: Typo in l. 329: rhidoid, not rhizoil

#####################################################################################################################################################################################################################################################

Reviewer #2: See attachment (if this isn't present, please immediately write me for it). It did not copy/paste accurately in this space.

7. PLOS authors have the option to publish the peer review history of their article (what does this mean?). If published, this will include your full peer review and any attached files.

Reviewer #1: No

Reviewer #2: No

---

## [Editor Report · Acceptance letter]

21 Mar 2024

PONE-D-23-16300R1 

PLOS ONE

Dear Dr. Hochroth, 

I'm pleased to inform you that your manuscript has been deemed suitable for publication in PLOS ONE. Congratulations! Your manuscript is now being handed over to our production team.

Kind regards, 

on behalf of

Dr. Frank Melzner 

Academic Editor

PLOS ONE